# Immune Evasion of SARS-CoV-2 Emerging Variants: What Have We Learnt So Far?

**DOI:** 10.3390/v13071192

**Published:** 2021-06-22

**Authors:** Ivana Lazarevic, Vera Pravica, Danijela Miljanovic, Maja Cupic

**Affiliations:** Institute of Microbiology and Immunology, Faculty of Medicine, University of Belgrade, 11000 Belgrade, Serbia; vera.pravica@med.bg.ac.rs (V.P.); danijela.karalic@med.bg.ac.rs (D.M.); maja.cupic@med.bg.ac.rs (M.C.)

**Keywords:** SARS-CoV-2, COVID-19, variant of concern, B.1.1.7, B.1.351, P.1, B.1.617, immune escape, mutations, neutralization

## Abstract

Despite the slow evolutionary rate of SARS-CoV-2 relative to other RNA viruses, its massive and rapid transmission during the COVID-19 pandemic has enabled it to acquire significant genetic diversity since it first entered the human population. This led to the emergence of numerous variants, some of them recently being labeled “variants of concern” (VOC), due to their potential impact on transmission, morbidity/mortality, and the evasion of neutralization by antibodies elicited by infection, vaccination, or therapeutic application. The potential to evade neutralization is the result of diversity of the target epitopes generated by the accumulation of mutations in the spike protein. While three globally recognized VOCs (Alpha or B.1.1.7, Beta or B.1.351, and Gamma or P.1) remain sensitive to neutralization albeit at reduced levels by the sera of convalescent individuals and recipients of several anti-COVID19 vaccines, the effect of spike variability is much more evident on the neutralization capacity of monoclonal antibodies. The newly recognized VOC Delta or lineage B.1.617.2, as well as locally accepted VOCs (Epsilon or B.1.427/29-US and B1.1.7 with the E484K-UK) are indicating the necessity of close monitoring of new variants on a global level. The VOCs characteristics, their mutational patterns, and the role mutations play in immune evasion are summarized in this review.

## 1. Introduction

Since the discovery of a pneumonia cluster of unknown origin in Wuhan province, China, in December 2019 [1], life on Earth has changed in a number of ways. The causative agent was identified as the severe acute respiratory syndrome coronavirus 2 (SARS-CoV-2), and soon, it became responsible for the pandemic of coronavirus disease 2019 (COVID-19). This severe respiratory syndrome has since led to the millions of infections and deaths worldwide as well setting to the test public health infrastructures and causing economic hardship.

Coronaviruses (CoVs), first isolated in 1962, were known as causative agents of mild respiratory and gastrointestinal infections in humans and animals [2,3]. However, the emergence of the severe acute respiratory syndrome coronavirus (SARS-CoV) in China in 2002 [4] and the Middle East respiratory syndrome coronavirus (MERS-CoV) in Saudi Arabia in 2012 [5] have changed the understanding of diseases caused by coronaviruses. These two viruses of zoonotic origin were highly pathogenic, causing fatal infections of the lower part of the respiratory tract [6]. The discovery of SARS-CoV-2 at the end of 2019 in China is considered to be the third jump of the coronaviruses from animals to humans. The high transmissibility of SARS-CoV-2 has led to the massive and rapid spread of the virus across the entire planet, and as of June 2021, more than 176 million people have been reported to be positive for SARS-CoV-2, more than 3.8 million died, and nearly 161 million recovered from COVID-19 [7]. The end of 2020 brought a glimmer of hope to this pandemic in the form of vaccination. Massive and rapid global vaccination, coupled with physical distancing, is the most effective method for resolving the pandemic in the long-term.

Although SARS-CoV-2 has a slow evolutionary rate compared to the other RNA viruses, massive and rapid transmission during pandemics has enabled it to acquire significant genetic diversity since it first entered the human population. This led to the emergence of variants that can potentially impact transmission, virulence, and antigenicity and that were since labeled internationally as variants of concern (VOCs). The characteristics of major, globally recognized VOCs, their mutational patterns, and the role that mutations play in immune evasion are summarized in this review.

## 2. Organization of SARS-CoV-2 Genome and Spike Protein

SARS-CoV-2 belongs to order *Nidovirales*, family *Coronaviridae,* subfamily *Orthocoronavirinae*, and genus *Betacoronavirus.* Virions of coronaviruses are spherical with average diameters of 80 to 120 nm. They are enveloped with positive single-stranded (ss) RNA genomes. The genomic analysis of three newly discovered coronaviruses showed that SARS-CoV-2 has 79% and 50% sequence similarity with SARS-CoV and MERS-CoV, respectively [8,9]. The coronavirus with the most similar genome to SARS-CoV-2 is horse-shoe bat virus RaTG13 *Rhinolophus affinis* with 96% of similarity [8,10]

SARS-CoV-2 genome is in the form of ssRNA with positive sense and a length of approximately 30,000 nucleotides. This non-segmented genome includes a 5′-untranslated region (*UTR*), followed by replicase complex (*ORF1a* and *ORF1ab*), structural genes for spike (S), envelope (E), membrane (M), nucleocapsid (N) proteins, and several open reading frames (ORFs) for accessory proteins inserted between four structural genes, ending with *3′-UTR* with poly A tail [1,11] (Figure 1). The *ORF1a* and *ORF1b* genes, located next to each other near *5′-UTR*, occupy two-thirds of SARS-CoV-2 genome and encode polyproteins pp1a and pp1ab. These two polyproteins are cleaved with autoproteolytic enzyme into 16 non-structural proteins (nsp1-16) that are involved in viral replication, transcription, immunomodulation, gene transactivation, and resistance to innate antiviral response [12]. The last third of the genome contains genes for structural and accessory proteins. The *S* gene encodes spike glycoprotein, which is the most prominent protein of the virion and enables viral entry into the target cell [13]. The M glycoprotein contains three domains, C terminal-, transmembrane-, and N terminal-domain, and it is necessary for the assembly and budding of virions [14]. The envelope protein also includes three domains and plays an important role in the pathogenesis of COVID-19 infection because its C-terminal domain binds to human tight junction protein PALS1 [15]. The nucleocapsid binds to viral RNA and influences the replication performance of SARS-CoV-2 [16]. Accessory proteins significantly contribute to evasion of the innate immune response by meddling with interferon (IFN) synthesis and obstructing signal pathways within the cell [17].

The spike is a transmembrane glycoprotein that is 1273 amino acids long and in the shape of a homotrimer. It comprises the receptor binding domain (RBD) that interacts with host cell receptor angiotensin converting enzyme 2 (ACE2) [18]. The SARS-Cov-2 S protein shares amino acid sequence similarity of 76.7–77% with SARS-CoVs from humans and civets, 75–98% with bat coronaviruses of the same subgenus (*Sarbecovirus*), and 90.7–92.6% with pangolin coronaviruses [8]. Spike includes two subunits: S1 (aa 1–685) and S2 (aa 686–1273) (Figure 2). The S1 subunit comprises an N-terminal domain (NTD) and RBD (aa 319–541), while the S2 subunit is composed of a fusion peptide (FP) (also called S2′ subunit), heptapeptide domain 1 and 2 (HPD1, HPD2), transmembrane domain (TM), and cytoplasm domain (CD) [19]. The RBD is the key player within the S1 subunit for the attachment of SARS-CoV-2 to ACE2 [20] and is therefore a very important target for antiviral drugs and antibodies [21]. It contains a core structure and receptor binding motif (RBM) (aa 437–508), which is the most variable part of spike protein that is important for binding to the outer surface of ACE2 [18]. There is only 73% similarity between RBD of SARS-CoV-2 and SARS-CoV, although they both bind to human ACE2 [22]. The characterization of the mechanical stability of the RBD of SARS-CoV-2 has shown it to be stiffer compared to SARS-CoV, which has important consequences in binding to ACE2, as it can withstand Brownian and cellular forces while remaining in close contact during the initial steps of cell entry [23]. The S protein persists in the open and closed form [24]. In the closed form, recognition motifs are hidden, and in the open form, the so-called receptor-accessible state, RBD with RBM are in up conformation. While in the up conformation, they stick out from the homotrimers and enable the binding to the receptor, fusion process, and virus entry into the host cell [20]. The fusion of two membranes is the crucial step in the viral life cycle [25]. It was suggested that certain key residues are crucial for stabilization of the spike protein during transitions from close to open conformations prior to ACE2 recognition [26]. Several high-frequency contacts are formed between the NTD and RBD that are responsible for the local conformational stability and play a role during the transition from closed to open state.

The S2 subunit has three conformational states: (1) pre-fusion native state; (2) pre-hairpin intermediate state; and (3) post-fusion hairpin state. The S1 and S2 subunits, which are non-covalently bound in the pre-fusion state, have a united role in binding to the receptor of the host cell [22]. The S1 subunit interacts with the receptor and enables attachment of the virion [27], while the S2 subunit is involved in the fusion process. The S protein is cleaved by host proteases, type II transmembrane serine proteases (TMPRSS2), and furin, at the junction of S1 and S2 at a polybasic cleavage site (S2′) [28,29]. The cleavage at the S2′ site activates the proteins, which induce irreversible conformational changes in the S protein, which is crucial for the fusion of viral and cell membranes [24]. The insertion of four amino acids (PRRA) at the polybasic cleavage site represents a specific genomic characteristic of SARS-CoV-2 [28]. This site is not observed in related coronaviruses except in a bat-derived coronavirus from *Rhinolophus malayanus* (RmYN02), which has an insertion of three amino acids (PAA) [30]. Some studies indicate that this furin-cleavage site induces the instability of SARS-CoV-2, causing conformational changes that are needed for the binding of RBD to the receptor [31].

## 3. Immune Response to SARS-CoV-2

Viruses, such as SARS-CoV-2, besides innate immunity, induce both humoral and cellular adaptive immune response, triggering different defense mechanisms in order to fight acute infection. NK cells, monocytes (macrophages), and IFN type I are crucial in the response to this virus. A fall in the number of NK cells [32] and plasmacytoid dendritic cells (main source of IFN type I) and dominating interleukin (IL)-6 producing monocytes are characteristic of inappropriate SARS-CoV-2 innate immune response [33]. https://pubmed.ncbi.nlm.nih.gov/33357409/Large (19 June 2021) trials, such as RECOVERY, established that anti-IL-6 in combination with steroids is a potential option for hypoxic patients with evidence of hyperinflammation [34]. There are data showing that a statistically significant correlation exists for some common variants in three genes linked to the innate immunity, MBL2, TMPRSS2, and CD27. MBL2 encodes a mannose-binding protein C that binds to mannose, activating the lectin complement pathway; TMPRSS2 cleaves the spike protein and ensures viral internalization [35], while the CD27 receptor is required for the generation and long-term maintenance of T-cell immunity. There are some data suggesting that trained innate immunity might also have a role in the protection against COVID-19 [36,37]. Several clinical trials are investigating whether unrelated vaccines, such as the measles, mumps, rubella vaccine, and the BCG vaccine can provoke trained innate immunity and improve protection against COVID-19 [38].

It is important to stress that recovery from COVID-19 infection is linked to appropriate immune response and disease severity is correlated to the impaired immune reaction. It is well known that this virus with its particular potential to inactivate the IFN-based response leads to the weakening of innate immunity. In addition, once present in the host cell, the SARS-CoV-2 activates the NOD-like receptor family, inducing the formation of an inflammasome. This contributes to the release of the pro-inflammatory cytokines, IL-1, IL-6, TNF, and IL-18. The NF-ĸB pathway is activated after interaction of the viral RNA with Toll-like receptors and enhances pro-inflammatory cytokines production. Thus, the inflammation starts and leads to the release of a number of cytokines from activated immune cells and the so-called cytokine storm, which can be life-threatening, happens [39].

Humoral immune response to SARS-CoV-2 is mediated by antibodies specific mainly to the spike glycoprotein, all of its parts including NTD, and the nucleocapsid protein [40]. These antibodies neutralize viral binding to cells expressing ACE2 receptors and infection of these cells [41]. Many studies that examine the duration of protection by functional neutralizing antibodies and the potential for re-infection have shown that most patients with COVID-19 have virus-specific IgM, IgA, and IgG responses in the days after infection [42]. In individuals with mild COVID-19, a rapid decline of RBD-specific IgG titers within 2–4 months has been observed in several studies, suggesting that SARS-CoV-2-induced humoral immunity might not be long-lasting in individuals with mild disease [43]. Antibody titers were significantly higher in patients with severe disease than in patients with mild disease and were associated with clinical outcomes [44]. However, a comprehensive study of adaptive immunity to SARS-CoV-2, which also examined the association with disease severity, showed that the concentration of neutralizing antibodies was not correlated with COVID-19 severity [45]. There is no pre-existing immunity to SARS-CoV-2 in the population, except through cross-reactivity with other coronaviruses [46]. It is also important to evaluate memory B-cells in addition to antibody measurement to better characterize humoral immunity. Although high circulating titers of neutralizing antibodies are common surrogates of protective immunity, there are many situations when circulating antibodies do not reach sufficient levels, and additional input from memory B-cells is necessary [47]. If circulating antibodies disappear over time, data suggest that robust memory B-cells are likely to provide a quick source of protective antibody in the case of potential SARS-CoV-2 re-infection. In addition, in infection with variants that can partially escape neutralization by present circulating antibodies [48,49,50], one will need vital memory B-cells to re-enter germinal centers and transform in order to respond to novel spike epitopes [51].

In addition, human monoclonal antibodies (mAbs) targeting both the NTD and RBD of SARS-CoV-2 have been isolated, with those targeting RBD being especially potent. These antibodies are used clinically [52,53], in therapeutic and prophylactic modes. Moreover, the selection of antibody mixtures with non-overlapping escape mutations should help and prolong the effectiveness of antibody therapies in SARS-CoV-2 infection [54].

Following the infection, a certain number of HLA-DR+ T-cells, both CD4+ and CD8+, rises in the first 7–10 days after the first symptoms and declines after three weeks [55,56,57]. The CD4+ T-cell response to SARS-CoV-2 predominantly consists of T-helper-1 (Th1) cells, which are characterized by high IFN-γ secretion and specificity for the structural spike glycoprotein, the membrane protein, and the nucleocapsid protein. CD8+ T-cell response specific to SARS-CoV-2 also produced IFN-γ and tumor necrosis factor (TNF). SARS-CoV-2-specific T-cells express perforin and granzymes after in vitro reactivation with viral antigens. It was also shown that during the convalescent phase, T-cells had a memory phenotype, both CD4+ and CD8+ T-cells expressing IFN-γ, IL-2, and TNF [58]. Response from T follicular helper (Tfh) cells is crucial to the development of strong humoral immunity through the formation of germinal centers and provision of co-stimulation (CD40–CD40-L interaction and cytokines) to B-cells [59]. A single-cell RNA sequencing study of the CD4+ T-cells specific to SARS-CoV-2 found an increased proportion of Tfh cells in patients with severe disease [60]. Other risk factors for severe COVID-19 are increased numbers of Th17 cells, T-cells expressing exhaustion markers (such as PD-1), and the depletion of both αβ and γδ T-cells [61]. The recognition of SARS-CoV-2 antigens by pre-existing and cross-reactive T-cells created during previous infection with human coronaviruses is also possible [62].

In the recent study [63], the authors have suggested that T-cell response and the binding of antibodies to the spike protein induce early protection in COVID-19. After mRNA vaccines, all individuals develop spike-specific T-cells, while 80% develop spike-binding antibodies 10 days after the first dose. They are suggesting that a lack of neutralizing antibodies is not essential to prevent against COVID-19. With the exception of killed whole-virus vaccines, all current vaccines offer S protein as the target immunogen, limiting T-cell immunity to spike epitopes. For the T-cell epitopes, a population exposure analysis proposed a set of epitopes that is estimated to provide broad coverage worldwide [64].

## 4. Genetic Variability of SARS-CoV-2 and Classification of Variants

The genetic diversity of SARS-CoV-2 is the result of errors generated by its RNA-dependent RNA polymerase (RdRp) and recombination [65]. The capacity of coronaviruses to recombine is associated with the strand switching ability of RdRp, and it is likely that it played a significant role in their evolution. Although coronaviruses have a slower mutation rate relative to other RNA viruses because of their proofreading 3′-to-5′ exoribonuclease (nsp14), the consequences of accumulating mutations are still a major concern. It became obvious that the accumulation of amino acid mutations might affect the transmissibility of the virus, its cell tropism, and its pathogenicity, presenting a serious challenge for the efficiency of current vaccines and diagnostic assays.

Until recently, the observed diversity among SARS-CoV-2 sequences has been low. The earliest spike protein mutation D614G of SARS-CoV-2 in Europe was identified in January 2020 in Germany [66]. Since then, the strain harboring D614G has become the dominant pandemic variant in most countries, possibly because the mutation enabled a relative fitness advantage to the original Wuhan strain and enhanced infectivity.

The accumulating number of SARS-CoV-2 variants has developed a need for their classification into groups such as lineages and clades. On 31 May 2021, the World Health Organization (WHO) introduced names based upon the Greek alphabet for important variants in order to simplify public communication around variants and enable referring to variants in a geographically neutral fashion [67]. However, this does not replace the three current nomenclature systems: GISAID (Global Initiative on Sharing All Influenza Data), Nextstrain, and PANGO. There are 8 clades of SARS-CoV-2 or hCoV-19 (S, O, L, V, G, GH, GR, and GV) identified by the GISAID database [68], 11 major clades (19A, 19B, and 20A–20I) recognized by Nextstrain, while Rambaut et al. [69] and the software team of the Phylogenetic Assignment of Named Global Outbreak Lineages (PANGOLIN) proposed 6 major lineages (A, B, B.1, B.1.1, B.1.177, B.1.1.7) now known as PANGO nomenclature. If a new, emerging variant possesses specific genetic markers that have been associated with increased transmissibility, morbidity and mortality, and ability to evade natural immunity as well as reduced neutralization by therapeutic antibodies or vaccination, reduced efficacy of treatments or potential diagnostic impact, it may be labeled “variant under investigation (VUI)” or “variant of interest (VOI)”, and if its prevalence and expansion surpasses the national level, it can be marked “variant of concern (VOC)”. If there is evidence that a variant has developed features that significantly reduce the effectiveness of existing prevention or intervention measures, it can be termed a “variant of high consequence” [70,71,72].

So far, there are four globally recognized variants of concern: Alpha or lineage B.1.1.7 (UK), Beta or lineage B.1.351 (South Africa), Gamma or lineage P.1 (Japan/Brazil), and Delta or lineage B.1.617.2 (India) [70,71,72,73]. Another was acknowledged as VOC in the UK and by ECDC—B.1.1.7 with E484K and two others by the US—Epsilon or B.1.427/29 [71,72,73].

Alpha or lineage B.1.1.7 (also known as 20I/501Y.V1 and VOC-202012/01) emerged in September 2020 in Southeastern England. It harbors seven missense mutations and three deleted residues in the spike protein [74]. Due to its enhanced transmissibility, it quickly spread worldwide and it is reported, as of 1 June 2021, in 160 countries. In February 2021, Public Health England (PHE) recognized B.1.1.7 with E484K mutation as a new VOC (VOC-202102/02), and it has since been identified in the US. However, this variant has not been detected in the UK since March 2021 but is continuing to spread outside the UK based on sequence data. Beta or lineage B.1.351 (also known as 20H/501Y.V2 variant) was first detected in the Eastern Cape province of South Africa in late 2020 [75]. It contains seven mutations and three deleted residues in spike protein. This variant is of the greatest concern in regard to immune escape for its three mutations within RBD and has since been spread to 113 countries. Variant Gamma or P.1 (also known as 20J/501Y.V3 variant) arising from lineage B.1.1.28 was first described in Brazil and Japan in December 2020 and later classified as VOC due to 11 spike mutations, including the same three in RBD as South African variant [76]. It has since been reported in 64 countries. Lineages B.1.429, defined by four and B.1.427 by two spike mutations are recognized as VOC in the US and as VOI Epsilon by WHO [77]. They were first identified in California (both also known as CAL.20C and 20C/S:452R), where they reached prevalence of more than 50% as of February 2021. As of June 2021, more than 60 countries reported cases caused by a newly recognized variant—lineage B.1.617 (also known as G/452R.V3) and its three sublineages, the first two detected in December 2020 and the third detected in February 2021 in India [70]. However, it has since become evident that only sublineage B.1.617.2 is associated with greater public health risk, which is why it is now the only sublineage of B.1.617 that is recognized as VOC—Delta [67]. Sublineage B.1.617.1 has been reclassified to a VOI (variant Kappa), and while it is still demonstrating increased transmissibility, global prevalence appears to be declining. Based upon reports, the prevalence of B.1.617.3 is low, and it is no longer classified as either a VOC or VOI.

The main speculation about the origin of novel variants with accumulated mutations is proposing that they evolved within immunosuppressed chronically infected patients who supported high viral replication for months and may have been treated with immune plasma or monoclonal antibodies [78,79,80]. However, since the lineages usually contain circulating intermediate mutants, the diversity within some lineages cannot be explained only by a single long-term infection in one individual [75].

## 5. Implications of SARS-CoV-2 Variants in Immune Evasion

Although different lineages are defined by mutations in more than one region of the genome, the most attention is paid to nonsynonymous changes in the S gene, which can alter the spike protein and influence its role in viral entry. This role of spike has determined it as an ideal target for immune response and also made it the primary target for most currently approved vaccines. Amino acid changes have been observed across the entire spike protein, but the exact location defines the impact of each substitution. The NTD and RBD are the most diverse regions, and most mAbs against SARS-CoV-2 that have been characterized target the RBM, and some are specific for RBD core- and NTD as well [81]. Changes in spike residues within major epitopes may reduce or ablate antibody binding and neutralization, which would lead to the diminished efficacy of antibodies, derived by natural infection or vaccination. However, changes are found to occur also within the conserved C-terminal domain of the S1 and the S2 subunit. These regions are important for conformational changes within S, which is needed for viral attachment and fusion, and may elicit still unknown neutralizing responses [82].

The first variant Alpha or B.1.1.7 that raised global concerns about increased transmissibility and potential immune evasion harbors seven missense mutations (N501Y, A570D, D614G, P681H, T716I, S982A, D1118H) and three deletions in spike (69/70del and 144del) (Figure 3). The three deleted residues are located within NTD, only one mutation (N501Y) is within RBM, three are displayed in the C-terminal domain (CTD) of S1, and three are displayed within S2. Various studies have so far demonstrated the reduced potency of neutralizing antibodies against B.1.1.7 [49,83,84,85,86,87,88]. These studies share the general conclusion that variant B.1.1.7 remains sensitive to neutralization, though at moderately reduced levels, by sera of convalescent individuals and recipients of several anti-COVID19 vaccines. The reduction in neutralization levels were on average 3-fold (ranging from 1.5-10-fold) for convalescent sera and ≈2-fold for sera of vaccine recipients (mRNA-, vector-, and subunit-based) [49,83,84,85,86]. However, when various mAbs were tested against this variant, it was uniformly shown that the B.1.1.7 variant can escape neutralization mediated by a fraction of RBM-specific antibodies and by most NTD-specific antibodies [49,83,88]. The proposed explanation for the more serious effect of spike mutations on neutralization by mAbs than by sera is the polyclonality of serum neutralization [84]. It is supported by the observation that a single mutation can diminish the binding of a single mAbs but not of other antibodies in the same binding cluster. A single mutation cannot affect all antibodies in the same cluster, since it seems that each antibody has well defined and unique molecular contact with the same specific epitope. Therefore, polyclonal sera are less susceptible to changes in neutralization due to a single mutation. Polyclonal sera also contain non-neutralizing antibodies whose role is yet to be elucidated.

The part of the diminished neutralizing effect of antibodies against the B.1.1.7 variant can be attributed to the only RBM mutation—N501Y. This mutation, shared by three globally recognized VOCs, is thought to be the result of viral adaptive evolution [89] and has been shown to increase affinity for ACE2 [85,90,91,92]. The enhanced binding affinity may be contributed to additional interactions with ACE2 that are allowed by 501 change—the new hydrogen bonds at residues 41 and 353 and also to a more open conformation of the RBD [93,94]. While some report its antigenic impact to be limited to a few mAbs with no significant effect on neutralization by convalescent or vaccinees sera [83], others show that the increase in transmission is combined with the reduction in the neutralization potency of convalescent sera [85]. The explanation for this lies not in the disrupted binding of antibodies to changed RBM but in competition of antibodies with ACE2 for binding to RBM. Thus, all changes in RBM that confer increased affinity for ACE2 will make the virus more difficult to neutralize.

The significant resistance of B.1.1.7 to neutralization by NTD-specific antibodies should be explained by the presence of three deleted residues in this region. The neutralization effect of these antibodies can be attributed to the role that NTD has in viral entry. While it was not yet determined for SARS-CoV-2, the NTD has a role in attachment to host cells in several CoV family members [95]. For SARS-CoV-2, it has been proposed that NTD interacts with auxiliary receptors in cell types that do not express ACE2 (e.g., DC-SIGN/L-SIGN) [96]. The NTD deletion H69/V70 is observed in B.1.1.7. and B.1.298 (Danish mink) but has not been associated so far with escape from NTD-specific antibodies [88]. A combination of del H69/V70 and N501Y was shown to increase infectivity in vitro [97]. On the other hand, deletion Y144 has been found to abrogate binding to neutralizing antibodies [49,52,88,98]. It can still not be determined whether NDT mutations are the result of immune selection or are generated as part of viral fitness improvement.

Other spike mutations of B.1.1.7 belong to the C-terminal domain of S1 and S2 and were not so far perceived to affect antibody neutralization. However, mutations within these regions might affect the conformation of RBD, attachment, and fusion, requiring further studies to determine their consequences and possible indirect effect on immune evasion. The extensively studied D614G was found to increase the ability of RBD to shift to the up position, which is necessary for interaction with ACE2 [99]. This resulted in the increased infectivity and transmissibility observed for the D614G variant relative to the original SARS-CoV-2 strains [100]. The P681H change is adjacent to the furin cleavage site and could potentially have an effect on S1/S2 cleavage and therefore on cell entry and infectivity.

The variant of the greatest concern in regard to immune escape, Beta or B.1.351, contains seven mutations (D80A, D215G, K417N, E484K, N501Y, D614G, A701V) and three deletions (241/242/243del) in the spike protein [75] (Figure 3). Two mutations (D80A, D215G) and three deleted residues are in the N-terminal domain of S1, one (A701V) is in loop 2 of S2 and 3 are at key residues in the RBD (K417N, E484K, N501Y). So far, there are multiple studies showing that B.1.351 decreases the neutralization capacity of antibodies elicited by infection with previous variants or vaccination [48,83,101,102,103,104,105]. This reduction in neutralizing potential for B.1.351 is most frequently detected in individuals with low antibody levels, and it is declining more rapidly with time [105], heightening concerns about re-infection or suboptimal protection by current vaccines. The problem in the non-vaccinated population exists because most people infected with SARS-CoV-2 develop only low to moderate titers, while higher titers are only observed in severely ill hospitalized individuals. The loss of neutralizing activity of convalescent plasma against B.1.351 ranged from 11 to 33-fold and by sera of vaccinees from 3.4 to 8.5-fold [50,83,101,103,104,105,106]. In addition, the B.1.351 variant showed resistance to neutralization by most NTD-specific and a number of RBM-specific mAbs [83,103,107].

The resistance to antibody neutralization of the B.1.351 variant is mainly ascribed to three mutations within RBD (K417N, E484K, N501Y). N501Y probably does not impair neutralization on its own but rather in combination with other two, which were found to partially compromise neutralization generated by previous infection or vaccination [48,103,106,107,108]. The result of the change at position 417 is loss of the polar interaction with residue D30 on human ACE2 [82]. However, a combination of K417N and N501Y was shown to enhance the binding with ACE2 and reduce binding with antibodies [109]. This improvement in receptor binding is supported by the observation of this mutation in a virulent mouse adapted strain of SARS-CoV-2 [110]. K417N was shown to be crucial to viral escape, effectively abrogating neutralization by some of the most common and potent neutralizing antibodies to SARS-CoV-2 [103]. Contrary to this, others [107] indicate that it may contribute to neutralization by enhancing the probability of conversion to the open conformation of the S protein, thus exposing epitopes to antibody neutralization.

Mutation E484K, which emerged independently in over 50 lineages, also corresponds with improved binding to ACE2. It enhances the binding affinity of N501Y for ACE2 still further but has been associated with immune escape from both mAbs and polyclonal sera as well [48,49,83,106,107]. Its location is within the RBD binding cleft, and it is considered to be a dominant neutralizing epitope [75,108,111]. The residue 484 can mutate into a diversity of different amino acids (E484A, E484G, E448D, and E484K) under the pressure of SARS-CoV-2 convalescent sera and exhibits resistance [112]. It is believed that the impact of mutation 484 on immune evasion is significantly augmented by the presence of other two RBD mutations in this variant, but its impact as the single point mutation was demonstrated as well [106,112].

The B1.1.7 variant bearing the E484K mutation emerged and was recognized as a variant of concern in the UK and Europe, since it appears to be responsible for a significant additional loss of neutralization capacity of monoclonal and polyclonal antibodies [49]. Monoclonal antibodies were shown to lose almost 50% of neutralizing activity against B.1.1.7 carrying E484K. A combination of E484K with various NTD mutations (particularly deletions) might prove to be even more effective in immune evasion [113], which is of the most significance in cases of both Beta variant and B1.1.7 with E484K.

The third globally recognized VOC, Gamma or P.1, is carrying 11 spike mutations. Five mutations are located within NTD (L18F, T20N, P26S, D138Y, R190S), three in RBD (K417T, E484K, N501Y), two in the C-terminal domain of S1 and near the furin cleavage site (D614G, H655Y), and one in S2 (T1027I) (Figure 3). Convalescent and vaccinee sera show a significant loss of neutralizing activity against P.1, but the reduction is not as substantial as against B.1.315 [114,115,116]. The loss of neutralizing activity of convalescent plasma against P.1 ranged from 6.5 to 13-fold and by sera of vaccinees from 2.2 to 2.8-fold [114,115], meaning that the neutralization of P.1 was not as severely compromised as that of B.1.351 and only slightly weakened compared to that of B.1.1.7. Not surprisingly, the neutralization activity of mAbs against P.1 is reduced much in the same manner as in B.1.351, since triple RBD mutations are mostly the same in both variants [114].

The reason for the differences in neutralization of B.1.351 and P.1 by the immune serum presumably reflects the difference in the mutations introduced outside the RBD. The role of NTD-specific neutralizing antibodies is not nearly yet defined. It was thought that extensive N-linked glycan shielding of NTD is diminishing its antigenicity, but in vitro studies showed the significant neutralizing capacity of some NTD-specific antibodies [52]. The fact that NTD is under selective pressure of human immune response is supported by the identification of NTD deletions in immunocompromised hosts with prolonged infections [79]. It is possible that neutralization assays based on target cells over-expressing ACE2 receptors are responsible for underrating the role of NDT-specific antibodies. Since NTD changes are much more distinct among three major VOC, it seems likely that neutralization variation among them is rather due to differences in NTD than RBD.

In January 2021, the emergence of a novel variant in California carrying an L452R mutation in the RBD was reported [77]. This variant (Epsilon) comprises two separate lineages B.1.427 and B.1.429, the first carrying two spike mutations (L452R, D614G) and the second carrying four (S13I, W152C, L452R, D614G). It is assumed that they emerged as early as May 2020 and they gained VOC status in the US due to significant increase in frequency from September 2020 to January 2021. In February 2021, they were identified in >50% of all sequenced cases in California and many other states [117]. They were shown to display moderate resistance to neutralization by convalescent sera (4–6.7-fold) and sera of vaccine recipients (2–2.9-fold) [48,117]. The RBD mutation L452R, shared by these lineages, is not located in the part that directly interacts with ACE2, but it is speculated that it may cause structural changes in the region that promote the interaction between the spike protein and its ACE2 receptor. Thus, the infectivity of pseudoviruses carrying L452R was shown to be higher than of the D614G variant but slightly reduced compared to that of N501Y variants [117]. The similar mechanism of RBD structural change due to L452R is offered in explanation of the reduced neutralization capacity of antibodies. This mutation, among several other RBD mutations, was selected by a panel of antibodies in vitro [112].

The emerging variant B.1.617 comprises three distinct sublineages (B.1.617.1, B.1.617.2, B.1.617.3) with different mutational profiles [70]. However, only sublineage B.1.617.2 or Delta is now internationally recognized as VOC. It is characterized by spike mutations T19R, G142D, Δ157-158, L452R, T478K, D614G, P681R, and D950N (Figure 3). The other two sublineages have a similar mutational profile: B.1.617.1 is defined by the spike amino acid changes G142D, E154K, L452R, E484Q, D614G, P681R, and Q1071H, and B.1.617.3 is defined by T19R, L452R, E484Q, D614G, P681R, and D950N. The presence of RBD mutations L452R, E484Q, and D614G in the C-terminal domain of S1 may result in the higher transmissibility of these sublineages due to their known impact on ACE2 binding and conformational changes important for ACE2 binding. All three sublineages of B.1.617 display P681R adjacent to the furin cleavage site and have enhanced S cleavage by furin, which is hypothesized to be enhancing transmissibility and pathogenicity [118]. Although the sublineage B.1.617.2 was initially considered to be as transmissible as B.1.1.7 [119], further evidence from the UK, based on the likelihood that close contacts of a person infected with the Delta variant will themselves become infected—the “secondary attack rate”, suggest that this variant may be over 60% more transmissible than the Alpha variant [120]. By recent report, more than 90% of new COVID-19 cases in the UK involve the Delta variant. The spread of the Delta variant is also registered in the US, where it now accounts for more than 6% of all infections (more than 18% of cases in some Western U.S. states) [121].

The impact on the immune escape capacity of three sublineages of B.1.617 is expected, owing to RBD mutations L452R, T478K, and E484Q and their combination with NTD mutations and deletions, particularly in the case of B.1.617.2. A similar change at position 478 (T478I) was previously selected in vitro and shown to exhibit reduced neutralization by monoclonal antibodies and human convalescent sera [112]. One of the first studies on B1.617.1 revealed that the neutralization capacity of convalescent sera and sera of recipients of inactivated killed vaccine was retained [122]. Other studies have reported a moderate reduction in neutralization of B1.617.1 by the sera of convalescents and recipients of mRNA vaccines and resistance to monoclonal antibodies approved for COVID-19 treatment [123,124,125]. The E484Q was found to have slightly milder impact but still corresponding to the effect of E484K, which is 10-fold reduction in the neutralization by sera of vaccine recipients. In addition, the combination of L452R and E484Q was not shown to have an additive effect; rather, the loss of sensitivity was similar to that observed with each mutation individually [124].

Finally, the impact of emerging SARS-CoV-2 variants of concern on cellular immune response should also be addressed in future research. It has been suggested that the resolution of SARS-CoV-2 infection and COVID-19 is significantly dependent on CD4+ and CD8+ T-cell responses [126], which also play a role in modulating disease severity [45,127]. In convalescent individuals, T-cell immunity is not restricted to spike-derived epitopes, and thus, it would be reasonable to assume that it would remain largely intact for new variants. However, in recipients of the majority of currently available vaccines, which offer S protein as the target immunogen, T-cell immunity is limited to spike epitopes. Therefore, it is of essence to determine whether new variant mutations in these epitopes impair T-cell responses in a similar way as escape from neutralizing antibodies. However, studies dealing with this problem are still scarce, mainly because measurements of T-cell immunity are more challenging for routine clinical practice than antibody detection assays. So far, the effect of variants B.1.1.7, B.1.351, P.1, and B.1.427/29 was found to be negligible on both CD4+ and CD8+ T-cell responses in the recipients of mRNA-based vaccines [128,129]. This was supported by the result of completely conserved epitopes for 93% of CD4+ and 97% of CD8+ T-cells in the variants. In addition, it was pointed out that HLA binding capacity is not affected in the majority of cases by a single mutation in epitopes. However, the repertoire of recognized epitopes is probably substantially different from one individual to the other due to HLA polymorphism, and thus, the negative impact of the mutations of specific variants on each single person could not be entirely dismissed [128].

## 6. Conclusions

As long as a significant number of the world population is infected with SARS-CoV-2, mutations will continue to occur because of the huge number of genome replications and error-prone replication. Therefore, new variants will continue to emerge, and some of them may pose a greater risk for immune escape. A selective pressure of adaptive immunity was minimal in the primarily naive world population, so most of the variants now present are the result of mutations derived by selection based on fitness advantage. The selective pressure for escape variants will probably increase as herd immunity is approached. In addition, the co-circulation of major variants in the same geographical region, already seen in many parts of the world, may enable recombination, bringing together mutations responsible for different consequences. Thus, the emergence of variants capable of immune evasion seems inevitable, and it will be important to evolve pandemic countermeasures accordingly.

While present research on the possible immune escape of emerging VOCs is still offering encouragement, there are several courses of action that can be undertaken in order to effectively subdue the pandemic. First, it would be necessary to closely monitor the emergence of novel SARS-CoV2 variants globally and to quickly recognize the potential for immune escape because it may well be possible that some of them are already present although still undetected. For full comprehension of immunity against novel variants, it would be essential to understand the immunogenicity of different spike domains as well as the role of non-neutralizing antibodies and cellular immunity. Vaccination protocols should be adjusted to always include two doses without delay of the second, since high neutralization titers could be crucial for protection against current variants. Finally, an effort should be made to modify currently used vaccines directed at ancestral spike and therapeutic protocols involving monoclonal antibodies in order to offer reliable protection against emerging variants.

## Figures and Tables

**Figure 1 viruses-13-01192-f001:**
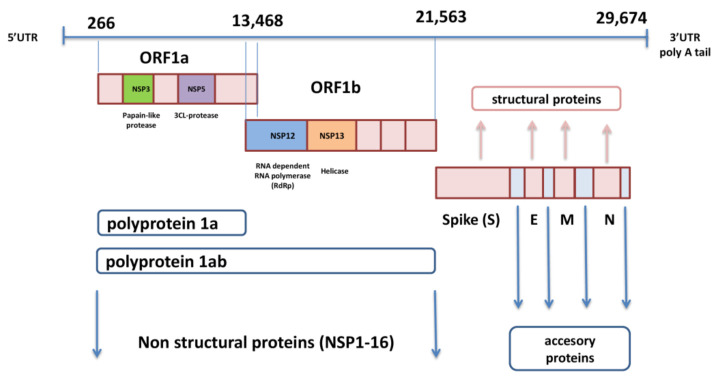
Organization of SARS-CoV-2 genome.

**Figure 2 viruses-13-01192-f002:**
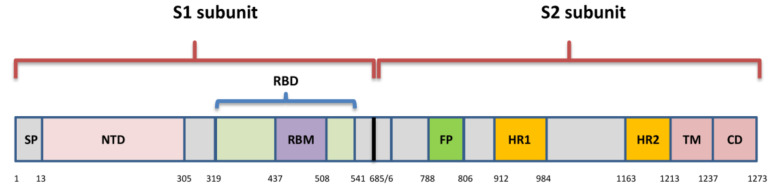
Spike protein of SARS-CoV-2; SP-signal peptide; NTD—N-terminal domain; RBD—receptor binding domain; RBM—receptor binding motif; FP—fusion peptide; HR1—heptapeptide domain 1; HP2—heptapeptide domain 2; TM—transmembrane domain; CD—cytoplasmic domain.

**Figure 3 viruses-13-01192-f003:**
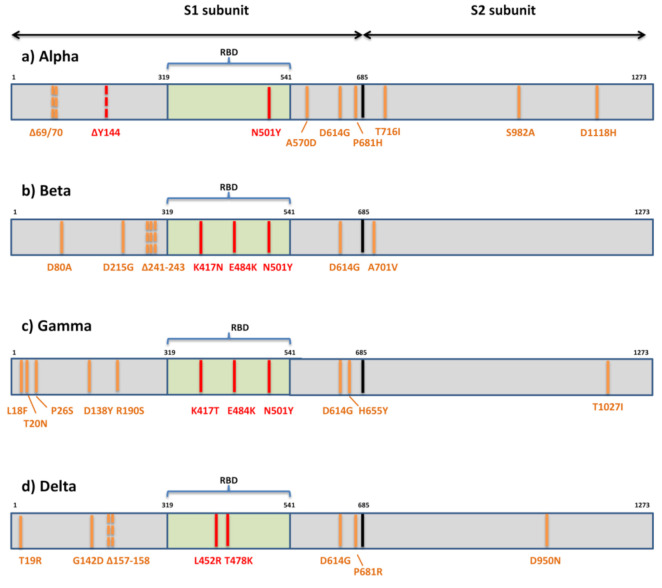
Mutational patterns of four variants of concern designated by the WHO. (**a**) Alpha or lineage B.1.1.7; (**b**) Beta or lineage B.1.351; (**c**) Gamma or lineage P.1; (**d**) Delta or lineage B.1.617.2; The residues potentially responsible for immune evasion are marked in red.

## Data Availability

The study did not report any new results or data.

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
