# Peer review of "Immune Evasion of SARS-CoV-2 Emerging Variants: What Have We Learnt So Far?"

_viruses, 2021, doi:10.3390/v13071192_

Round 1

Reviewer 1 Report

The manuscript “Immune evasion of SARS-CoV-2 emerging variants: what have we learnt so far?” by Lazarevic and colleagues describes a summary of what is known about several of the main variants-of-concern (VOCs) that are currently going around. First, the basic biology of SARS-CoV-2 is described, followed by a short summary of what kind of immune response SARS-CoV-2 elicit. Subsequently, the authors dive into 4 main variants, B.1.1.7 (alpha), B.1.351 (beta), P.1 (gamma), and B.1.617.2 (delta) as well as the B.1.427/9 California variants, focusing how specific mutations in the Spike protein might or might not interfere with antibody binding. This is a very timely topic, as we are now about 1.5 year into the COVID pandemic and new information is reported on a daily basis. The rise of new variants has also caused significant concerns in both a global and regional manner. The delta variant is thought to be the primary variant for the latest and deadliest peak in India, and now is now spreading at concerning spread in the UK and US. The gamma variant is thought to be the primary variant spreading and maintaining a very high infection rate in South America. These two variants were preceded by the alpha variant which quickly spread from the UK around the world. Recent high infection numbers in South-East Asia, a better understanding of the biology of the variants is essential. Overall, it is a clear manuscript with a few minor concerns, which are listed below.

Minor concerns:

+ Recently the nomenclature (alpha, beta, etc) has changed for the variants. To stay up to date, it would be ideal if the latest nomenclature is used, both in the text and figures.

+ RBM on page 3, line 97 is not explained.

+ On page 5, lines 219-223, the authors state that the mutation D614G had relative fitness advantage to the WT strain. Although this is a very reasonable hypothesis, because this mutation spread across the world which was naïve to virus. It is therefore very difficult to discern if the efficient spread of D614G was due to relative fitness advantage or founder effect. This reviewer suggests to be a little more cautious in this paragraph.

+ Ideally Figure 3 and 4 are merged so it is easier for the reader to follow which residues are different. Also, if the residues which might be functional are highlighted, this would also help the reader to more quickly understand what your message is.

+ On page 10, line 441, the authors state that the delta variant is as transmissible as the alpha variant. But a recent reports (https://www.biorxiv.org/content/10.1101/2021.05.28.446163v1 and https://assets.publishing.service.gov.uk/government/uploads/system/uploads/attachment_data/file/991343/Variants_of_Concern_VOC_Technical_Briefing_14.pdf) from the UK suggests that the delta variant is 50-66% more transmissible than the alpha variant, as its increased spreading in the UK and US are supportive of this report.

+ The one thing this reviewer was missing in this review is brief discussion of how the variants are going around in the current hotspots. Maybe it is worth very briefly touching on this.

+ There are a few spellings, grammar, and formatting errors, such as nucelocapsid on page 3, line 82.

Author Response

Point 1: Recently the nomenclature (alpha, beta, etc) has changed for the variants. To stay up to date, it would be ideal if the latest nomenclature is used, both in the text and figures.

Response 1: The latest nomenclature was introduced in both text and figures.

Point 2: RBM on page 3, line 97 is not explained

Response 2: The corrections were made so that all abbreviations are now explained.

Point 3: On page 5, lines 219-223, the authors state that the mutation D614G had relative fitness advantage to the WT strain. Although this is a very reasonable hypothesis, because this mutation spread across the world which was naïve to virus. It is therefore very difficult to discern if the efficient spread of D614G was due to relative fitness advantage or founder effect. This reviewer suggests to be a little more cautious in this paragraph.

Response 3: These sentences were rephrased in order to leave all possibilities open.

Point 4: Ideally Figure 3 and 4 are merged so it is easier for the reader to follow which residues are different. Also, if the residues which might be functional are highlighted, this would also help the reader to more quickly understand what your message is.

Response 4: Figure 3 and 4 are merged and sublineages B.1.617.1 and B.1.617.3 are omitted since they are no longer considered variants of concern. The residues potentially responsible for immune evasion are marked in red.

Point 5: On page 10, line 441, the authors state that the delta variant is as transmissible as the alpha variant. But a recent reports (https://www.biorxiv.org/content/10.1101/2021.05.28.446163v1 and https://assets.publishing.service.gov.uk/government/uploads/system/uploads/attachment_data/file/991343/Variants_of_Concern_VOC_Technical_Briefing_14.pdf) from the UK suggests that the delta variant is 50-66% more transmissible than the alpha variant, as its increased spreading in the UK and US are supportive of this report.

Response 5: This was updated and the latest data from Public Health England on transmissibility of Delta variant is included.

Point 6: The one thing this reviewer was missing in this review is brief discussion of how the variants are going around in the current hotspots. Maybe it is worth very briefly touching on this.

Response 6: The recent data on prevalence of variants are added, particularly for the Delta variant which has now become dominant in the UK and is rapidly spreading in the US.

Point 7: There are a few spellings, grammar, and formatting errors, such as nucelocapsid on page 3, line 82.

Response 7: Spelling, grammar and formatting errors are corrected.

Reviewer 2 Report

The review by Lazarevic is a very good compendium of the current VOCs. It is nice to read and above all, it cites extensively studies on the spike protein and the mutations which are concerning the world. If I may suggest,  in the introduction one can comment on:

- I believe the authors do comment very briefly the fundaments of mutation in SARS-CoV-2 S protein and its functional consequence (i.e. high binding and resistant to AB). But, I would like to call the attention to the effect of mutations in the spike protein by also altering the mechanical stability (https://doi.org/10.1039/D0NR03969A) of the SARS-CoV-2 over the SARS (2002-2003) and the enegetics (https://doi.org/10.3390/ma13235362) of the closed to open conformation..

Author Response

Point 1: I believe the authors do comment very briefly the fundaments of mutation in SARS-CoV-2 S protein and its functional consequence (i.e. high binding and resistant to AB). But, I would like to call the attention to the effect of mutations in the spike protein by also altering the mechanical stability (https://doi.org/10.1039/D0NR03969A) of the SARS-CoV-2 over the SARS (2002-2003) and the enegetics (https://doi.org/10.3390/ma13235362) of the closed to open conformation..

Response 1: Sentences are added to the paragraph about organization of SARS-CoV-2 genome and spike protein in order to address these issues.